# FILTER-RECOVERY NETWORK FOR MULTI-SPEAKER AUDIO-VISUAL SPEECH SEPARATION

**Haoyue Cheng** [1,2]    **Zhaoyang Liu** [2]    **Wayne Wu** [2]    **Limin Wang** [1, 2, *]

[1]State Key Laboratory for Novel Software Technology, Nanjing University, China
[2]Shanghai AI Laboratory
{chenghaoyue98,zyliumy,wuwenyan0503}@gmail.com   lmwang@nju.edu.cn

## ABSTRACT

In this paper, we systematically study the audio-visual speech separation task in a multi-speaker scenario. Given the facial information of each speaker, the goal of this task is to separate the corresponding speech from the mixed speech. The existing works are designed for speech separation in a controlled setting with a fixed number of speakers (mostly 2 or 3 speakers), which seems to be impractical for real applications. As a result, we try to utilize a single model to separate the voices with a variable number of speakers. Based on the observation, there are two prominent issues for multi-speaker separation: 1) There are some noisy voice pieces belonging to other speakers in the separation results; 2) Part of the target speech is missing after separation. Accordingly, we propose **BFRNet**, including a **B**asic audio-visual speech separator and a Filter-Recovery Network (**FRNet**). FR-Net can refine the coarse audio separated by basic audio-visual speech separator. To have fair comparisons, we build a comprehensive benchmark for multi-speaker audio-visual speech separation to verify the performance of various methods. Experimental results show that our method is able to achieve the state-of-the-art performance. Furthermore, we also find that FRNet can boost the performance of other off-the-shelf speech separators, which exhibits its ability of generalization.

## 1 INTRODUCTION

Audio-visual speech separation has been extensively used in various applications, such as speech recognition (Radford et al.; Chan et al., 2015), assistive hearing device (Kumar et al., 2022), and online video meetings (Tamm et al., 2022). As human voices are naturally mixed together in public places, it would be challenging to directly extract the information of interest from such raw audio-visual signals containing multiple speakers. As a result, separating audio signals for each speaker could serve as an effective pre-processing step for further analysis on the audio-visual signal.

Convolutional neural networks (Gogate et al., 2018; Makishima et al., 2021; Gao & Grauman, 2021) and Transformers (Ramesh et al., 2021; Montesinos et al., 2022; Rahimi et al., 2022) has made prominent progress in the field of audio-visual speech separation. However, previous works (Lee et al., 2021; Gao & Grauman, 2021; Montesinos et al., 2022) mostly focus on two-speaker speech separation. Although other researches (Ephrat et al., 2018; Afouras et al., 2018b; 2019) manage to separate voices for more speakers, (Ephrat et al., 2018) requires customized models for each kind of mixture instead of separating all kinds of mixtures with a single model, and (Afouras et al., 2018b) mainly contributes to enhancing the voice of the target individual while ignoring others. Furthermore, (Afouras et al., 2019) uses pre-enrolled speaker embeddings to extract the corresponding speech, but still leaves a performance gap compared with unenrolled speakers.

Therefore, how to efficiently and effectively separate speech under a multi-speaker environment still requires further study. Through our explorations, prior works Gao & Grauman (2021); Montesinos et al. (2022); Chuang et al. (2020); Makishima et al. (2021); Afouras et al. (2019) show their superiority in two-speaker speech separation but yield disappointing results with more speakers (*e.g.*, 3, 4, or even 5 speakers). It exhibits a simple yet important fact that the complexity of speech separation

---

*Corresponding author.

strongly correlates with the number of speakers in the mixed audio. As shown in Fig. 1, the core problems for multi-speaker speech separation can be empirically summarized as two folds:

1) The noisy parts (red box in Figure 1) contains the components from other speakers within the separated audio . 2) The missing parts (gray box in Figure 1) represent some target speech pieces dropped by models. These two problems usually occur in complex conditions, reflecting that models fail to accurately separate audio signals under such challenging scenarios.

As a result, there is still a long way to go toward solving audio-visual speech separation for multi-speaker conditions.

To this end, we come up with an effective method BFRNet, to conquer the aforementioned challenges

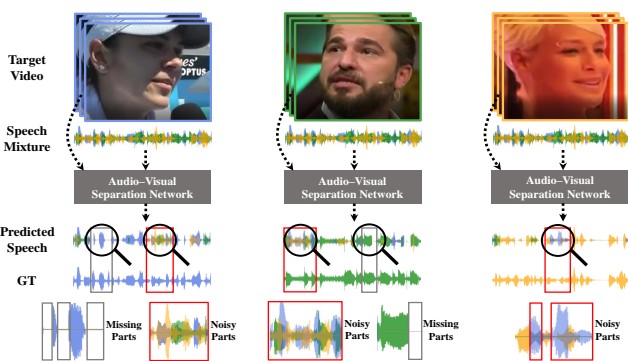

Figure 1: **Illustration of two critical issues in multi-speaker audio-visual speech separation.** The missing parts in separation results compared to ground truth speech are indicated by gray boxes, while the noisy parts are marked by red boxes.

in multi-speaker audio-visual speech separation. As illustrated in Figure 2, the BFRNet consists of a basic audio-visual speech separator and a Filter-Recovery Network (FRNet). The FRNet aims at solving the two issues in the results of any identically structured basic separator. It comprises two serial modules, *i.e.*, Filter and Recovery. Firstly, the Filter module utilizes the visual features of the target speaker to query corresponding audio signals from the coarse prediction and suppress components from other speakers. Then, the Recovery module uses the clean audio yielded by the Filter module to query the missing components that belong to the target speaker from others' predictions. Essentially, FRNet aims to calibrate the coarsely separated audio predicted by off-the-shelf models (Gao & Grauman, 2021; Montesinos et al., 2022; Chuang et al., 2020).

Furthermore, we have noticed that there is still one obstacle in evaluating audio-visual speech separation. As we found that most works (Gao & Grauman, 2021; Montesinos et al., 2022) evaluate performance on unfixed numbers of samples generated by randomly mixing the audio signals during each inference, it sometimes leads to hard reproductions and unfair comparisons. Consequently, to unify the evaluation protocols, we create a comprehensive benchmark to verify the models' performance fairly. Specifically, for each type of mixture, we randomly sample test videos without replacement to make up the speech mixtures. The constructed fixed test sets serve for all experiments to ensure fairness and reproducibility of the results.

To sum up, our contributions are as follows:

- *First*, we design a Filter-Recovery Network (FRNet) with multi-speaker training paradigm to improve the quality of multi-speaker speech separation.
- *Second*, to test the different methods on a fair basis, a well-established benchmark for multi-speaker audio-visual speech separation is created. We not only unify the evaluation protocol but also re-implement several state-of-the-art methods on this benchmark.
- *Finally*, in the experiments, we demonstrate that our proposed FRNet can be equipped with other models to further improve the quality of audio separation and achieve the state-of-the-art performance.

## 2    RELATED WORKS

**Audio-Only Speech Separation.** Using only audio modality for speech separation faces the problem of speaker agnosticism. Some works (Liu et al., 2019; Wang et al., 2018) utilize the speaker's voice embedding as the hint to isolate the target speech. Current methods mostly treat the audio-only speech separation as a label permutation problem. (Chen et al., 2017) cluster the similar speech to perform speech separation. (Luo et al., 2018) requires no prior information of speaker number. (Luo

& Mesgarani, 2019) adopts a deep learning network comprising a series of convolutional layers and trains the network with permutation-invariant loss (Yu et al., 2017).

**Visual Sound Separation.** There are various kinds of sound separation explored in the literature. One type focuses on music separation (Zhao et al., 2018; Gao & Grauman, 2019; Xu et al., 2019; Gan et al., 2020), where the diverse shapes of musical instruments and the distinguished patterns of music sounds are the key clues. Other works concentrate on the in-the-wild sounds (Gao et al., 2018; Tzinis et al., 2020; Chen et al., 2020), such as animal sounds, vehicle sounds, etc. (Gao et al., 2018) incorporate the image recognition results as the advisor and learn prototypical spectral patterns for each sounding object. (Tzinis et al., 2020) extend to unsupervised, open-domain audio-visual sound separation and develop a large new dataset. (Chen et al., 2020) propose an algorithm to generate temporally synchronized sound given mismatched visual information.

**Visual Speech Separation.** Speech is extensively studied as the most closely associated audio with humans, and has a broad range of applications. Due to the natural correlation between face and speech, many works (Gabbay et al., 2018; Lu et al., 2018; Ephrat et al., 2018; Afouras et al., 2019; Chung et al., 2020; Hegde et al., 2021; Rahimi et al., 2022) employ face-related information to separate speech from a mixture in the literature. (Chung et al., 2020) use only the still images containing facial appearance to isolate speech, with the assistance of the consistency of face identity and speech identity. Numerous methods (Afouras et al., 2018b; Gao & Grauman, 2021; Ephrat et al., 2018) explore the simultaneous lip motions and voice fluctuations clues. (Lu et al., 2018) integrate optical flow and lip movements to predict the spectrogram masks. (Hegde et al., 2021) propose synthesizing a virtual visual stream to deal with the situation where the visual stream is unreliable or completely absent. Another family of works (Owens & Efros, 2018; Afouras et al., 2020; Truong et al., 2021) combines multiple tasks for joint learning.

The most relevant works to ours are (Afouras et al., 2018b), (Shi et al., 2020), and (Yao et al., 2022). (Afouras et al., 2018b) also pays attention to the uncontrolled environment with several speakers, but it only focuses on enhancing the target speech and suppressing the noisy voices. Our method further separates every speech component for different mixtures in a single model. (Shi et al., 2020) and (Yao et al., 2022) perform a coarse-to-fine procedure by adopting an additional refining separation phase. However, the refining phase applies the same model as the coarse phase. Thus if the coarse phase cannot achieve a clean separation, the refining results will likewise be sub-optimal.

## 3 METHOD

To best of our knowledge, this is the first work that systematically studies the audio-visual speech separation under a multi-speaker setting. In this section, we first elaborate on our proposed multi-speaker training strategy in Sec. 3.1; Next, we introduce the adopted basic audio-visual speech separator in Sec. 3.2, which takes the visual information and the speech mixture as input and outputs separated speech; Then the Filter-Recovery Network is detailedly described in Sec. 3.3; Finally, we formulate the objective function for training model in Sec. 3.4.

### 3.1 OVERVIEW

Given a video containing $S$ simultaneous speakers, our goal is to isolate the individual speech for each speaker. Formally, we denote time-domain speech mixture as $x = \sum_{i=1}^{S} x_i, x_i \in \mathbb{R}^{T_x}$, where $T_x$ represents the time length, and $x_i$ is the separate speech of the $i\text{-}th$ speaker. Since acquiring the exact individual ground truth data from the mixtures in real scenes is yet impossible, we follow previous works (Afouras et al., 2018b; Gao & Grauman, 2021) to synthesize mixtures by adding individual speech together.

Current works are designed for definite speakers in the mixture, mostly 2 or 3 speakers. However, these models present poor effects for practical applications, where there are usually a variable number of speakers. Alternatively, they take an inefficient strategy of training separate models for each kind of mixture different in speaker numbers. To meet the demand for practical applications, we create different mixtures with various numbers of speakers during training, i.e., $S$ ranges from 2 to 5. The network is required to isolate all speech components for all kinds of mixtures.

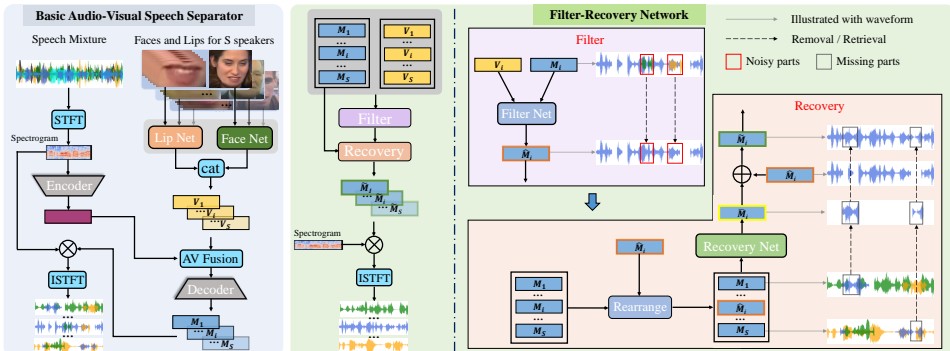

Figure 2: **Overview of the proposed framework.** It consists of a basic audio-visual speech separator and a Filter-Recovery Network (FRNet). Given a mixture spectrogram containing $S$ pieces of speech, the basic separator takes each speaker's visual clue separately and outputs the corresponding visual feature and speech mask. For $i$-$th$ speaker, the basic separator outputs $V_i$ and $M_i$. Then the visual features and speech masks of all $S$ speakers in a mixture are fed into the Filter-Recovery Network to obtain a more precise speech mask $\tilde{M}_i$ for each speaker. The Filter Net utilizes visual embedding $V_i$ to reduce the noisy components in $M_i$ to obtain $\hat{M}_i$. Then the Recovery Net takes the clean mask $\hat{M}_i$ as the query to extract the corresponding missing parts from the speech masks of other speakers. To better illustrate the effect of the FRNet, we visualize masks into the form of waveforms to present speech components by $\rightarrow$. The red/gray boxes and the corresponding $\dashrightarrow$ represent the noisy/missing speech pieces that are removed or retrieved.

Fig. 2 illustrates the architecture of the proposed model, consisting of a basic audio-visual speech separator and an additional Filter-Recovery Network (FRNet). During training, we feed mixtures into the audio-visual speech separator and obtain the separated coarse speech. The model takes the time-frequency spectrogram $X \in \mathbb{R}^{2 \times F \times T_X}$ as input, which is converted from time-domain audio $x$ via Short-Time Fourier Transform (STFT). $F$ and $T_X$ indicate the spectrogram's maximum frequency and time span. Each time-frequency bin contains the real and imaginary parts of the complex spectrogram. For $i$-$th$ speaker, the basic separator outputs target mask $M_i \in \mathbb{R}^{2 \times F \times T_X}$ of input spectrogram $X$ and visual feature $V_i$.

Afterward, for each mixture, the separated $S$ masks and visual features are sent to the FRNet to filter the noisy components and recover the missing ones. The FRNet outputs an improved mask for each speaker in the mixture, which is finally used to restore the exact time-domain speech.

## 3.2 BASIC AUDIO-VISUAL SPEECH SEPARATOR

In our framework, the basic audio-visual speech separator can be replaced with any network that outputs the masks of the mixture spectrogram. Following (Gao & Grauman, 2021), we adopt both lip and face clues to guide target speech separation. Specifically, a lip net is taken to dig out the consistency of lip motion and continuous pronunciation, and a face net is used to explore the relationship between speech and face attributes.

**Lip Net.** Following the previous structure (Ma et al., 2021; Gao & Grauman, 2021), we feed $T_v$ consecutive frames of lip regions into a 3D convolutional layer followed by a ShuffleNet v2 (Ma et al., 2018) network to extract mouth features. A temporal convolutional network is further utilized to output the lip motion features $Lip_i \in \mathbb{R}^{C_l \times T_v}$ for $i$-$th$ speaker.

**Face Net.** The Face Net aims at leveraging the correspondence between face attributes and speech. A ResNet-18 network takes a single face image as input, and outputs a face embedding for $i$-$th$ speaker. We repeat the face embedding along the time dimension to obtain $Face_i \in \mathbb{R}^{C_f \times T_v}$.

**Encoder-Decoder Separator.** As for the speech analysis and separation end, we adopt a U-Net network consisting of an encoder, a fusion module, and a decoder. The encoder is composed of multiple convolutions and pooling layers. It takes the mixture spectrogram $X$ as input and outputs an audio feature $Au$ of dimension $C_a \times T_v$. Following (Xiong et al., 2022), we take an AV-Fusion module, an attention-like operation, to obtain vision-related audio representation for a given speaker. We first concatenate the lip feature $Lip_i$ and face feature $Face_i$ together along the time dimension

to get visual feature $V_i \in \mathbb{R}^{C_v \times T_v}$, then feed $V_i$ and $Au$ to AV-Fusion module to obtain an enhanced feature. It is further delivered into the decoder to predict a mask $M_i$ of input spectrogram for $i$-$th$ speaker, which denotes the projection values of the prediction onto the mixture. Note that the values might be positive or negative, as the spectrograms have both positive and negative values.

## 3.3 Filter and Recovery Network

Due to the high complexity of speech mixture with multiple speakers (especially above 3 speakers), the separated target speech remains two apparent issues: 1) There still exists voices from others; 2) The isolated speech is partially missing compared to the ground truth speech. To solve the two problems, we introduce the Filter-Recovery Network (FRNet), which takes the visual features and the separated coarse speech masks as input, and outputs more precise target speech masks. For convenience, we concatenate the real and imaginary parts of $M_i$ obtained by the basic separator to form dimension $2F \times T_X$ before feeding it into FRNet, which we still denote $M_i$. Besides, to align the dimension, we transform $V_i$ to $V_i'$ of shape $2F \times T_X$ with a convolutional layer.

As Fig. 2 shows, the FRNet consists of a Filter Net and a Recovery Net. The former utilizes visual features to remove the noisy voices from the separated speech, and the latter learns the correlation between all separated speech to extract the missing voices from others.

Since the attention mechanism (Vaswani et al., 2017) enhances some parts of the input data while declining other parts, we adopt attention as the component module. For the sake of fluent description, we define some operations here. Given three tensors $\mathbf{q} \in \mathbb{R}^{D_h \times N_q}$, $\mathbf{k}, \mathbf{v} \in \mathbb{R}^{D_h \times N_{kv}}$, we compute the weighted sum of $\mathbf{v}$:

$$[\mathbf{q}', \mathbf{k}', \mathbf{v}'] = [\mathbf{U}_q \mathbf{q}, \mathbf{U}_k \mathbf{k}, \mathbf{U}_v \mathbf{v}], \qquad \mathbf{U}_{q,k,v} \in \mathbb{R}^{D_h \times D_h}, \qquad (1)$$

$$W = \mathrm{softmax}\left((\mathbf{q}')^\top \mathbf{k}'/\sqrt{D_h}\right), \qquad W \in \mathbb{R}^{N_q \times N_{kv}}, \qquad (2)$$

$$\mathrm{Attn}(\mathbf{q}, \mathbf{k}, \mathbf{v}) = \mathbf{v}' W^\top, \qquad \mathrm{Attn} \in \mathbb{R}^{D_h \times N_q}. \qquad (3)$$

**Filter Net.** Since the visual information is highly correlated with speech, we utilize each speaker's visual knowledge to reduce irrelevant voices in the separation results during the Filter phase. The Filter Net consists of $L$ basic layers, each consisting of an $\mathrm{Attn}$ and an MLP module, where the MLP block contains two fully-convolutional layers and a ReLU activation function.

For $l$-$th$ layer, the model takes $V_i'$ and $M_i^l$ as input, and outputs $M_i^{l+1}$, where $M_i^0 = M_i$, and LN denotes Layer Normalization (Ba et al., 2016):

$$z^{l+1} = \mathrm{LN}\left(\mathrm{Attn}(V_i', M_i^l, M_i^l) + M_i^l\right), \qquad z^{l+1} \in \mathbb{R}^{2F \times T_x}, \qquad (4)$$

$$M_i^{l+1} = \mathrm{LN}\left(\mathrm{MLP}(z^{l+1}) + z^{l+1}\right), \qquad M_i^{l+1} \in \mathbb{R}^{2F \times T_x}. \qquad (5)$$

For $i$-$th$ speaker, the Filter Net output mask $\hat{M}_i = M_i^L$ where the noise speech pieces are removed.

**Recovery Net.** According to our analysis, there are some pieces of speech from other speakers in the target separation result of basic separator. So we design the Recovery Net to pull out the missing voice pieces $\bar{M}_i$ for $i$-$th$ speaker from the separation results of other speakers, which is much easier than recovering the missing parts from the original mixture. We define a rearrange operation by stacking $S - 1$ masks of other speakers:

$$\mathrm{temp} = [M_1; \cdots; M_{i-1}; M_{i+1}; \cdots; M_S], \quad \mathbf{M}_i \xleftarrow{\mathrm{reshape}} \mathrm{temp}, \quad \mathbf{M}_i \in \mathbb{R}^{T_X \times 2F \times (S-1)}, \quad (6)$$

where the $\xleftarrow{\mathrm{reshape}}$ operation rearranges the input tensor to the target dimension.

The Recovery Net aims to learn the association between $\hat{M}_i$ and $\mathbf{M}_i$. It consists of $L$ basic layers, similar to the decoder of Transformer (Vaswani et al., 2017). Specifically, the output of layer $l + 1$ can be conveyed by the following equations, where $q_i^0 = \hat{M}_i$:

$$\hat{q}_i^l = \mathrm{LN}(\mathrm{Attn}(q_i^l, q_i^l, q_i^l) + q_i^l), \qquad \bar{q}_i^l \xleftarrow{\mathrm{reshape}} \hat{q}_i^l, \qquad \hat{q}_i^l \in \mathbb{R}^{2F \times T_X}, \bar{q}_i^l \in \mathbb{R}^{T_X \times 2F \times 1}, \quad (7)$$

$$z^l[t] = \mathrm{LN}\left(\mathrm{Attn}\left(\bar{q}_i^l[t], \mathbf{M}_i[t], \mathbf{M}_i[t]\right) + \bar{q}_i^l[t]\right), \qquad z^l[t] \in \mathbb{R}^{2F \times 1}, \qquad (8)$$

$$\hat{z}^l[t] = \mathrm{LN}\left(\mathrm{MLP}(z^l[t]) + z^l[t]\right), \qquad \hat{z}^l[t] \in \mathbb{R}^{2F \times 1}, \qquad (9)$$

$$\hat{z}^l = [\hat{z}^l[0]; ...; \hat{z}^l[t]; ...; \hat{z}^l[T_X]], \quad q_i^{l+1} \xleftarrow{\mathrm{reshape}} \hat{z}^l, \quad \hat{z}^l \in \mathbb{R}^{T_X \times 2F \times 1}, q_i^{l+1} \in \mathbb{R}^{2F \times T_X}. \quad (10)$$

We hold the view that the speech separation issues analyzed above do not always manifest to the same degree at different time slots. As a result, in Equ. 8, the attention operations are performed separately for each time slice of tensor to recover missing parts. Finally, the missing components $\bar{M}_i$ can be obtained by applying a fully-convolutional layer on $q_i^L$, i.e., $\bar{M}_i = \text{FC}(q_i^L)$.

After obtaining the noise-filtered result $\hat{M}_i$ and the recovered missing part $\bar{M}_i$, we add them up to obtain the clean and complete separation target mask:

$$\tilde{M}_i = \hat{M}_i + \bar{M}_i. \tag{11}$$

### 3.4 Loss Function

We jointly optimize the outputs of the basic separator and the FRNet. Following previous work (Pan et al., 2022), We take the scale-invariant signal-to-noise ratio (SI-SNR) (Le Roux et al., 2019) as the loss function. The predicted masks $M_i$ and $\tilde{M}_i$ are separately multiplied by the mixture spectrogram $X$ to get the separated spectrograms, which are finally transformed by inverse STFT to restore the time-domain speech $y_i$ and $\tilde{y}_i$. Given any prediction $\hat{s}$ and ground truth $s$, the SI-SNR loss can be computed with the following formula:

$$\mathcal{L}_{SI\text{-}SNR}(\hat{s}, s) = -10 \log_{10}\left(\frac{\|\frac{\langle \hat{s}, s\rangle s}{\|s\|^2}\|^2}{\|\hat{s} - \frac{\langle \hat{s}, s\rangle s}{\|s\|^2}\|^2}\right). \tag{12}$$

We jointly train the basic separator and the FRNet by an overall loss for $k$-$th$ speaker:

$$\mathcal{L}_i = \lambda \mathcal{L}_{SI\text{-}SNR}(y_i, x_i) + (1 - \lambda)\mathcal{L}_{SI\text{-}SNR}(\tilde{y}_i, x_i), \tag{13}$$

where $\lambda$ is the factor to control the ratio of the two loss parts. For a batch of mixtures containing a total number of $N$ speakers, the training loss is the average of all $N$ individual losses.

## 4 Experiments

### 4.1 Datasets

**VoxCeleb2** (Chung et al., 2018). This dataset is organized in the identity labels, with 5994 speakers in the training set and another 118 in the test set. It contains more than 1 million samples, each consisting of an utterance and synchronized face tracks. Following (Gao & Grauman, 2021), we hold out two videos for each speaker in the original training set and utilize the rest videos as our training set. For the remaining videos in the original training set, we randomly sample 7200 videos to build the seen test set. The speakers also appear in the training set, but the specific utterances do not. Our unseen test set consists of 7200 videos randomly chosen from the original test set. All of the rest videos in the VoxCeleb2 dataset form our validation set.

**Lip Reading Sentences 2&3** (Afouras et al., 2018a;c). LRS2 and LRS3 datasets contain 144k and 151k video clips from TV programs. To evaluate the generalization of models, we train them on the VoxCeleb2 dataset and test them on cross-domain LRS2 and LRS3 datasets without fine-tuning. For each dataset, the test set consists of 1200 randomly selected videos from the original test set.

Table 1: **Numbers of videos and mixtures for four test sets.**

| Test Dataset | # Videos | # 2-mix | # 3-mix | # 4-mix | # 5-mix |
|---|---|---|---|---|---|
| VoxCeleb2 unseen set | 7200 | 3600 | 2400 | 1800 | 1440 |
| VoxCeleb2 seen set | 7200 | 3600 | 2400 | 1800 | 1440 |
| LRS2 | 1200 | 600 | 400 | 300 | 240 |
| LRS3 | 1200 | 600 | 400 | 300 | 240 |

**Test Benchmark**. During the test, we thoroughly evaluate the models' performance on mixtures with various numbers of speakers, i.e., from mixtures with 2 speakers to 5 speakers. Meanwhile, to eliminate the uncertainty of results caused by testing on randomly created mixtures each time, we construct fixed test sets of mixtures and perform all experiments on these predetermined mixtures.

For each dataset, we randomly mix the test videos to build 2-mix, 3-mix, 4-mix, and 5-mix test sets without any subjective considerations. Each video appears only once in each mixture set. To be more explicit, we list the number of videos and each kind of mixture for all test sets in Tab. 1.

## 4.2 IMPLEMENTATION DETAILS

**Data Process**. Following the previous setting (Gao & Grauman, 2021), we randomly cut 2.55-second long clips from videos sampled at 25fps and the corresponding audios sampled at 16kHz as the training pairs. For all utterances that make up a mixture, we first normalize the energy of each one to the same, which corresponds to the same loudness for each utterance. Then we add up all normalized speech to obtain the mixture. STFT is conducted on the mixture waveform using a Hann window length of 400, a hop size of 160, and an FFT window size of 512 to output the complex spectrogram of dimension $2 \times 257 \times 256$, which is taken as the input to the U-Net encoder. A randomly selected frame from the video is rescaled to $224 \times 224$ and sent to the face analysis network. The input to the lip reading network is 64 consecutive frames of cropped gray mouth regions of dimension $88 \times 88$. We adopt the official implementation of 2D face landmark detection (Bulat & Tzimiropoulos, 2017) to detect the mouth landmarks and crop the mouth regions.

**Training Setting**. To achieve great separation results for mixtures containing different numbers of speakers, we separate all types of mixtures into individual speech simultaneously. In each batch, the ratio of 2-mix, 3-mix, 4-mix, and 5-mix numbers is set to 2:1:1:1, and the total number of speakers is 256. As for selecting utterances to create mixtures, some methods (Gao & Grauman, 2021; Rahimi et al., 2022) perform put-back random sampling. Such an approach results in some utterances being sampled multiple times while others are ignored, in which case the model might overfit some samples and underfit others. Instead, at the beginning of each training epoch, we shuffle all samples randomly and pick them one by one to ensure no duplication or omission.

**Optimization.** Empirically, we adopt the Adam optimizer to train the network with a weight decay of 1e-4 and a learning rate of 1e-4. We drop the learning rate by a factor of 0.1 after epochs 12 and 15, and train models for 19 epochs, when the loss almost reaches a plain. Tab. 8 displays experimental results of setting different lambda values, and we finally set it to 0.5.

**Evaluation.** Following previous methods (Afouras et al., 2018b; Ephrat et al., 2018; Gao & Grauman, 2021; Rahimi et al., 2022), we adopt the standard blind source separation metric Signal-to-Distortion-Ratio (SDR) (Vincent et al., 2006), which measures the ratio between the energy of the target signal and that of the errors. To further assess the speech quality and intelligibility, we also employ the Perceptual Evaluation of Speech Quality (PESQ) (Rix et al., 2001) metric.

## 4.3 COMPARISON WITH STATE-OF-THE-ART METHODS

Tab. 2 and 3 compare our method to other open-source state-of-the-art models on the test sets. All experiments follow the same test and evaluation protocols. We choose two types of methods for comparison: audio-only methods and audio-visual methods.

**Audio-Only Methods**:

*U-Net-AO*. By removing the face net and the lip net from the adopted basic audio-visual speech separator and making corresponding adaptations for the U-Net, we obtain an audio-only speech separator and train it with permutation-invariant loss.

*VoiceFilter* (Wang et al., 2018). This method uses the speech mixture and a reference utterance to extract the target speech from the mixture. Note that the reference audio is a piece of a randomly sampled utterance of the target speaker different from the target speech.

*Conv-TasNet* (Luo & Mesgarani, 2019). This method is widely adopted in audio-only speech separation practices. It adopts a fully-convolutional audio separation network, which takes the time-domain speech mixture as input and outputs all speech components simultaneously.

**Audio-Visual Methods**:

*LAVSE* (Chuang et al., 2020). To reduce processing costs, this work employs a lightweight but efficient framework, where a lip encoder extracts the lip motion feature as the synchronization signal for target speech extraction.

*VisualVoice* (Gao & Grauman, 2021). It contains the same visual networks and U-Net as our basic audio-visual separator, except the U-Net simply concatenates the audio and visual features, while we perform AV-Fusion to obtain enhanced audio features and achieve better results on VoxCeleb2. Be-

Table 2: **Results on VoxCeleb2 dataset.** The metrics are the average of all speakers for each test set. All methods are trained in the proposed multi-speaker setting. BFRNet achieves the best performance among all methods for all types of mixtures. (e.g., we achieve **11.06** dB SDR on VoxCeleb2 unseen 2-mix set.) Besides, we list the overall (O.A.) performance of all methods for convenient comparison, which is the average performance of each speaker in all kinds of mixtures. It is worth mentioning that each audio-visual method combined with the FRNet results in a large improvement.

| Method / # Spk. | Unseen | | | | | | | | | | Seen | | | | | | | | | |
|---|---|---|---|---|---|---|---|---|---|---|---|---|---|---|---|---|---|---|---|---|
| | SDR (dB) | | | | | PESQ | | | | | SDR (dB) | | | | | PESQ | | | | |
| | 2 | 3 | 4 | 5 | O.A. | 2 | 3 | 4 | 5 | O.A. | 2 | 3 | 4 | 5 | O.A. | 2 | 3 | 4 | 5 | O.A. |
| **Audio-Only** | | | | | | | | | | | | | | | | | | | | |
| U-Net-AO | 0.06 | -2.93 | -4.66 | -5.87 | -3.35 | 1.55 | 1.15 | 0.98 | 0.89 | 1.14 | 0.06 | -2.92 | -4.65 | -5.87 | -3.34 | 1.59 | 1.17 | 1.01 | 0.94 | 1.18 |
| VoiceFilter | 7.07 | 4.21 | 1.98 | -0.57 | 3.17 | 2.39 | 2.08 | 1.87 | 1.22 | 1.89 | 7.56 | 4.53 | 2.12 | 0.13 | 3.59 | 2.54 | 2.27 | 1.99 | 1.53 | 2.08 |
| Conv-TasNet | 9.46 | 4.57 | 1.50 | -0.62 | 3.73 | 2.67 | 2.11 | 1.78 | 1.66 | 2.05 | 9.88 | 4.76 | 1.54 | -0.65 | 3.88 | 2.70 | 2.13 | 1.79 | 1.66 | 2.07 |
| **Audio-Visual** | | | | | | | | | | | | | | | | | | | | |
| LAVSE | 5.38 | 1.53 | -0.6 | -2.13 | 1.05 | 2.17 | 1.67 | 1.43 | 1.28 | 1.64 | 5.54 | 1.45 | -0.72 | -2.31 | 0.99 | 2.20 | 1.67 | 1.43 | 1.28 | 1.64 |
| LAVSE + FRNet | 8.45 | 4.63 | 2.25 | 0.45 | 3.95 | 2.57 | 2.07 | 1.79 | 1.58 | 2.00 | 8.70 | 4.66 | 2.16 | 0.24 | 3.94 | 2.59 | 2.07 | 1.77 | 1.56 | 2.00 |
| VisualVoice | 8.97 | 4.59 | 2.05 | 0.25 | 3.96 | 2.64 | 2.12 | 1.82 | 1.62 | 2.05 | 8.61 | 3.96 | 1.39 | -0.4 | 3.39 | 2.61 | 2.05 | 1.74 | 1.54 | 1.98 |
| VisualVoice + FRNet | 10.78 | 7.04 | 4.62 | 2.76 | 6.30 | 2.86 | 2.43 | 2.13 | 1.91 | 2.33 | 10.95 | 6.95 | 4.37 | 2.40 | 6.17 | 2.87 | 2.41 | 2.09 | 1.86 | 2.31 |
| VoViT | 9.62 | 5.08 | 2.06 | -0.21 | 4.14 | 2.66 | 2.12 | 1.77 | 1.53 | 2.02 | 9.91 | 5.29 | 2.17 | -0.11 | 4.31 | 2.69 | 2.15 | 1.80 | 1.54 | 2.04 |
| VoViT + FRNet | 10.72 | 6.74 | 3.82 | 1.44 | 5.68 | 2.76 | 2.27 | 1.92 | 1.65 | 2.15 | 11.05 | 6.97 | 3.98 | 1.55 | 5.89 | 2.79 | 2.30 | 1.95 | 1.68 | 2.18 |
| DeBaSe | 10.08 | 5.91 | 3.16 | 1.06 | 5.05 | 2.72 | 2.23 | 1.90 | 1.65 | 2.12 | 10.05 | 5.44 | 2.51 | 0.28 | 4.57 | 2.72 | 2.17 | 1.82 | 1.56 | 2.07 |
| BFRNet (ours) | **11.06** | **7.48** | **5.13** | **3.26** | **6.73** | **2.87** | **2.48** | **2.20** | **1.97** | **2.38** | **11.27** | **7.48** | **4.89** | **2.86** | **6.63** | **2.89** | **2.47** | **2.16** | **1.91** | **2.36** |

Table 3: **Results on LRS 2&3 datasets.** To validate the generalization of separation models, they are trained on VoxCeleb2 and validated on LRS2/LRS3 without fine-tuning. BFRNet outperforms all base methods that do not integrate FRNet. Although some methods combined with FRNet achieve the best performance in certain metrics, it still proves the effectiveness of FRNet. Similarly, we give the overall (O.A.) performance on all kinds of mixtures for each test set for direct comparison.

| Method / # Spk. | LRS2 | | | | | | | | | | LRS3 | | | | | | | | | |
|---|---|---|---|---|---|---|---|---|---|---|---|---|---|---|---|---|---|---|---|---|
| | SDR(dB) | | | | | PESQ | | | | | SDR(dB) | | | | | PESQ | | | | |
| | 2 | 3 | 4 | 5 | O.A. | 2 | 3 | 4 | 5 | O.A. | 2 | 3 | 4 | 5 | O.A. | 2 | 3 | 4 | 5 | O.A. |
| **Audio-Only** | | | | | | | | | | | | | | | | | | | | |
| U-Net-AO | 0.25 | -2.65 | -4.26 | -5.37 | -3.01 | 1.51 | 1.14 | 0.97 | 0.90 | 1.13 | 0.19 | -2.70 | -4.34 | -5.45 | -3.08 | 1.40 | 1.03 | 0.85 | 0.76 | 1.01 |
| VoiceFilter | 6.98 | 4.10 | 1.78 | -0.77 | 3.02 | 2.35 | 2.01 | 1.69 | 1.17 | 1.80 | 8.02 | 4.58 | 2.95 | -0.31 | 3.81 | 2.48 | 2.08 | 1.86 | 1.26 | 1.92 |
| Conv-TasNet | 9.26 | 4.16 | 1.13 | -1.10 | 3.36 | 2.48 | 1.88 | 1.51 | 1.30 | 1.79 | 10.35 | 4.68 | 3.45 | -0.58 | 4.48 | 2.64 | 1.98 | 1.79 | 1.38 | 1.95 |
| **Audio-Visual** | | | | | | | | | | | | | | | | | | | | |
| LAVSE | 5.74 | 1.68 | -0.54 | -2.16 | 1.18 | 2.04 | 1.54 | 1.30 | 1.17 | 1.51 | 6.10 | 1.80 | -0.39 | -1.96 | 1.39 | 2.05 | 1.50 | 1.26 | 1.10 | 1.48 |
| LAVSE + FRNet | 8.85 | 4.58 | 1.92 | -0.06 | 3.82 | 2.45 | 1.90 | 1.58 | 1.37 | 1.83 | 9.46 | 4.78 | 2.11 | 0.04 | 4.10 | 2.53 | 1.91 | 1.58 | 1.35 | 1.84 |
| VisualVoice | 10.31 | 5.55 | 2.62 | 0.56 | 4.76 | 2.61 | 2.04 | 1.68 | 1.47 | 1.95 | 11.21 | 6.09 | 3.20 | 1.04 | 5.38 | 2.72 | 2.09 | 1.73 | 1.47 | 2.00 |
| VisualVoice + FRNet | **11.70** | **7.83** | **5.02** | **2.90** | **6.86** | **2.79** | **2.35** | **1.99** | **1.73** | **2.22** | 12.66 | 8.47 | 5.73 | 3.54 | 7.60 | 2.93 | 2.45 | 2.09 | 1.79 | 2.32 |
| VoViT | 10.03 | 4.66 | 1.15 | -1.48 | 3.59 | 2.56 | 1.94 | 1.55 | 1.29 | 1.84 | 10.64 | 5.37 | 1.41 | -0.88 | 4.14 | 2.64 | 1.97 | 1.51 | 1.26 | 1.84 |
| VoViT + FRNet | 11.29 | 6.50 | 2.84 | -0.07 | 5.14 | 2.67 | 2.10 | 1.67 | 1.40 | 1.96 | 11.92 | 7.10 | 3.35 | 0.70 | 5.77 | 2.75 | 2.17 | 1.70 | 1.40 | 2.01 |
| DeBaSe | 9.95 | 4.62 | 1.09 | -1.10 | 3.64 | 2.54 | 1.91 | 1.50 | 1.25 | 1.8 | 11.25 | 5.80 | 2.55 | -0.22 | 4.85 | 2.69 | 2.04 | 1.63 | 1.30 | 1.92 |
| BFRNet (ours) | 11.68 | 7.37 | 4.15 | 1.86 | 6.26 | 2.78 | 2.28 | 1.86 | 1.58 | 2.12 | **12.74** | 8.37 | 5.38 | 2.80 | 7.32 | 2.92 | 2.42 | 2.03 | 1.69 | 2.26 |

sides, VisualVoice contains an additional vocal analysis network designed for only 2-mix separation, so we remove the vocal analysis network to adapt to our multi-speaker setting.

*VoViT* (Montesinos et al., 2022). The newly proposed method uses a landmark-based graph convolutional network (Yan et al., 2018) to capture the facial motion cues. It adopts an AV spectro-temporal transformer for target speech separation.

*DeBaSe*. In order to offset the impact of increased model capacity brought by the additional FRNet, we compare the results of a deeper basic audio-visual speech separator (DeBaSe) to that of BFRNet, which has more layers in the encoder and has almost the same parameter counts as BFRNet.

Besides, we combine all audio-visual methods apart from DeBaSe with the proposed FRNet to verify its generality. We call these methods '*+FR'.

As seen in Tab. 2, Tab. 3 and Fig. 3, BFRNet achieves the best results in all test sets compared to other baseline methods with at least 1 dB SDR ad-

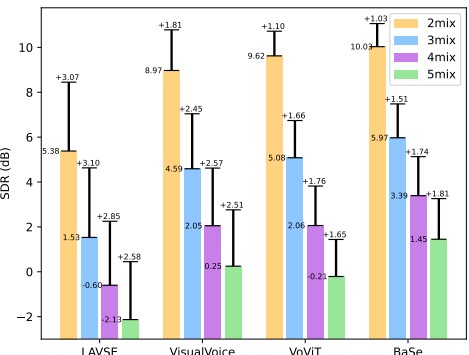

Figure 3: **Visualization of SDR for each method on VoxCeleb2 unseen test set.** The increments on each bar denote the improvement after combining FRNet with base methods. 'BaSe' is our basic audio-visual separator.

vantage. Besides, the FRNet combined with any audio-visual method can significantly improve performance. For instance, VisualVoice (Gao & Grauman, 2021) combined with FRNet receives a 1.67 dB gain of SDR on the VoxCeleb2 unseen 2-mix test set. It is worth noting that there is still a large gap between DeBaSe and BFRNet, although their parameter amounts are nearly equal. It means that just expanding the model capacity does not necessarily lead to significant performance gain. In contrast, BFRNet focuses on the main issues of the separation task, thus significantly enhancing the separation results. The difficulty of separation grows with the increase of speaker number in the mixture, resulting in lower metrics. For the VoxCeleb2 dataset, although the speakers in the seen test set appear in the training set, the results of the seen set are not certainly higher than that of the unseen set. We argue that the speakers' voiceprint features is not very critical for speech separation.

## 4.4 ABLATION STUDIES

We conduct ablation studies and report average results on VoxCeleb2 unseen and seen test sets.

**The effect of visual knowledge.** We remove the Face Net and Lip Net separately while remaining other parts. Tab. 4 shows that modified models consistently yield lower results than BFRNet. However, as lip motion is more related to speech, it plays a principal role in speech separation, while the static face image only serves as extra information.

**The effect of FRNet modules.** To validate the necessity of each module, we conduct experiments in Tab. 5 that remove individual modules. When the Filter net is removed, the Recovery net adopts $M_i$ to replace $\hat{M}_i$. When the Recovery net is removed, the output of Filter net $\hat{M}_i$ is considered the final output.

**Module design of FRNet.** Tab. 6 explores deformed structures of FRNet. For Filter Net, we filter voice noise from $M_i$ with itself as a clue instead of a simultaneous visual signal $V_i$. We name this ablation 'filter-sa'. As to Recovery Net, we replace the clean mask $\hat{M}_i$ with $M_i$ as the clue to extract the missing voices from $M_i$, and we name it 'recovery-noise'.

Table 4: **Experiments on effect of visual clues.**

| Modules | | SDR (dB) | | | | PESQ | | | |
|---|---|---|---|---|---|---|---|---|---|
| Face Net | Lip Net | 2 | 3 | 4 | 5 | 2 | 3 | 4 | 5 |
| | ✓ | 11.07 | 7.43 | 4.99 | 3.06 | 2.83 | 2.44 | 2.14 | 1.92 |
| ✓ | | 8.25 | 4.40 | 1.28 | -1.39 | 2.54 | 1.90 | 1.53 | 1.31 |
| ✓ | ✓ | 11.17 | 7.48 | 5.01 | 3.06 | 2.88 | 2.48 | 2.18 | 1.94 |

Table 5: **Ablation study on FRNet modules.**

| Modules | | SDR (dB) | | | | PESQ | | | |
|---|---|---|---|---|---|---|---|---|---|
| Filter | Recovery | 2 | 3 | 4 | 5 | 2 | 3 | 4 | 5 |
| | | 10.05 | 5.79 | 3.12 | 1.13 | 2.74 | 2.24 | 1.93 | 1.70 |
| | ✓ | 10.8 | 6.65 | 4.51 | 2.48 | 2.83 | 2.38 | 2.10 | 1.86 |
| ✓ | | 10.3 | 6.30 | 3.75 | 1.81 | 2.85 | 2.37 | 2.05 | 1.82 |
| ✓ | ✓ | 11.17 | 7.48 | 5.01 | 3.06 | 2.88 | 2.48 | 2.18 | 1.94 |

Table 6: **Ablation study on module design.**

| Exps | SDR (dB) | | | | PESQ | | | |
|---|---|---|---|---|---|---|---|---|
| | 2 | 3 | 4 | 5 | 2 | 3 | 4 | 5 |
| Filter-sa | 10.91 | 7.19 | 4.81 | 2.86 | 2.85 | 2.37 | 2.86 | 1.82 |
| Recovery-noise | 10.81 | 7.21 | 4.82 | 2.88 | 2.86 | 2.43 | 2.12 | 1.88 |
| FR | 11.17 | 7.48 | 5.01 | 3.06 | 2.88 | 2.48 | 2.18 | 1.94 |

Table 7: **Ablation study on the layers $L$.**

| Layers | SDR (dB) | | | | PESQ | | | |
|---|---|---|---|---|---|---|---|---|
| | 2 | 3 | 4 | 5 | 2 | 3 | 4 | 5 |
| 1 | 10.88 | 7.17 | 4.69 | 2.74 | 2.83 | 2.41 | 2.10 | 1.86 |
| 2 | 11.17 | 7.48 | 5.01 | 3.06 | 2.88 | 2.48 | 2.18 | 1.94 |
| 3 | 11.20 | 7.52 | 5.17 | 3.31 | 2.88 | 2.50 | 2.18 | 1.95 |

**Layers of the FRNet.** Both the Filter and Recovery Nets are composed of $L$ basic layers. We here study the impact of $L$. As seen in Tab. 7, there is a great improvement for the network with $L = 2$ compared to $L = 1$. When the number of layers $L$ increases to 3, the performance improvement is insignificant. To balance performance and efficiency, we adopt the 2-layer FRNet.

## 4.5 QUALITATIVE RESULTS

We visualize the intensity of ground truth (GT) spectrograms and predictions by models in Fig 4. The first row presents the ground truth, the separation result of the basic separator, and the result of BFRNet, respectively. The second row shows the separation results of other approaches. The results demonstrate that the separation results of baseline methods are subject to the two issues we claim, including the results of the proposed method without the FRNet. We use black boxes to highlight the missing parts compared to GT, and red boxes to indicate the noisy parts from other speakers. As seen in Fig. 4, the two problems are greatly suppressed after utilizing FRNet.

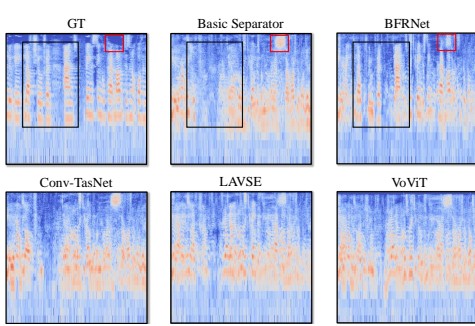

Figure 4: **Spectrogram visualization of ground truth and predictions.**

## 5 CONCLUSION

In this paper, we have focused on the multi-speaker audio-visual speech separation task. We are the first to propose separating mixtures with a variable number of speakers simultaneously during training, and we also provide a standard test benchmark for a fair comparison. There are two significant problems for speech separation, especially in the multi-speaker setting: part of the voice is missing in the separated speech; the separated speech may still be mixed with others' voices. To deal with this, we propose a Filter and Recovery network to solve these two problems. The filter module filters out other people's voices, and the recovery module compensates for their missing voices. We conduct various experiments to demonstrate the effectiveness of this module, and its addition to other audio-visual speech separation methods has led to considerable improvements.

## ACKNOWLEDGEMENTS

This work is supported by the National Key R&D Program of China (No. 2022ZD0160900), the National Natural Science Foundation of China (No. 62076119, No. 61921006), the Fundamental Research Funds for the Central Universities (No. 020214380091), and the Collaborative Innovation Center of Novel Software Technology and Industrialization.

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

# A  APPENDIX

We provide an additional ablation study and more visualization examples in the appendix.

**Ablation study on $\lambda$ in loss function.** We conduct experiments to explore the impact of different training $\lambda$ in Eq. 13 on the results. A smaller $\lambda$ means a higher training loss weight of FRNet. On the contrary, a larger $\lambda$ implies a lower weight of FRNet in the whole model. Tab. 8 displays the average results

Table 8: **Ablation Study on $\lambda$.**

| $\lambda$ | SDR (dB) | | | | PESQ | | | |
|---|---|---|---|---|---|---|---|---|
| | 2 | 3 | 4 | 5 | 2 | 3 | 4 | 5 |
| 0.2 | 11.13 | 7.47 | 5.01 | 3.08 | 2.86 | 2.47 | 2.19 | 1.95 |
| 0.5 | 11.17 | 7.48 | 5.01 | 3.06 | 2.88 | 2.48 | 2.18 | 1.94 |
| 0.8 | 10.99 | 7.20 | 4.63 | 2.54 | 2.84 | 2.41 | 2.09 | 1.83 |

of VoxCeleb2 unseen and seen test sets. As seen in the table, attaching a higher weight to the training of FRNet ($\lambda$ is 0.2) yields similar results to giving an equal weight ($\lambda$ is 0.5). Nevertheless, a lower loss weight of FRNet ($\lambda$ is 0.8) results in a non-negligible performance drop, which proves the necessity of FRNet.

**Spectrogram visualization of BFRNet.** We visualize the intensity of spectrograms of ground truth (GT) and predictions by each network of BFRNet in Fig. 5. The red boxes indicate the noisy part generated by the basic separator and then suppressed by Filter Net. The black boxes denote the missing part yielded by the basic separator and further recovered by Recovery Net. The results demonstrate the effects of Filter Net and Recovery Net.

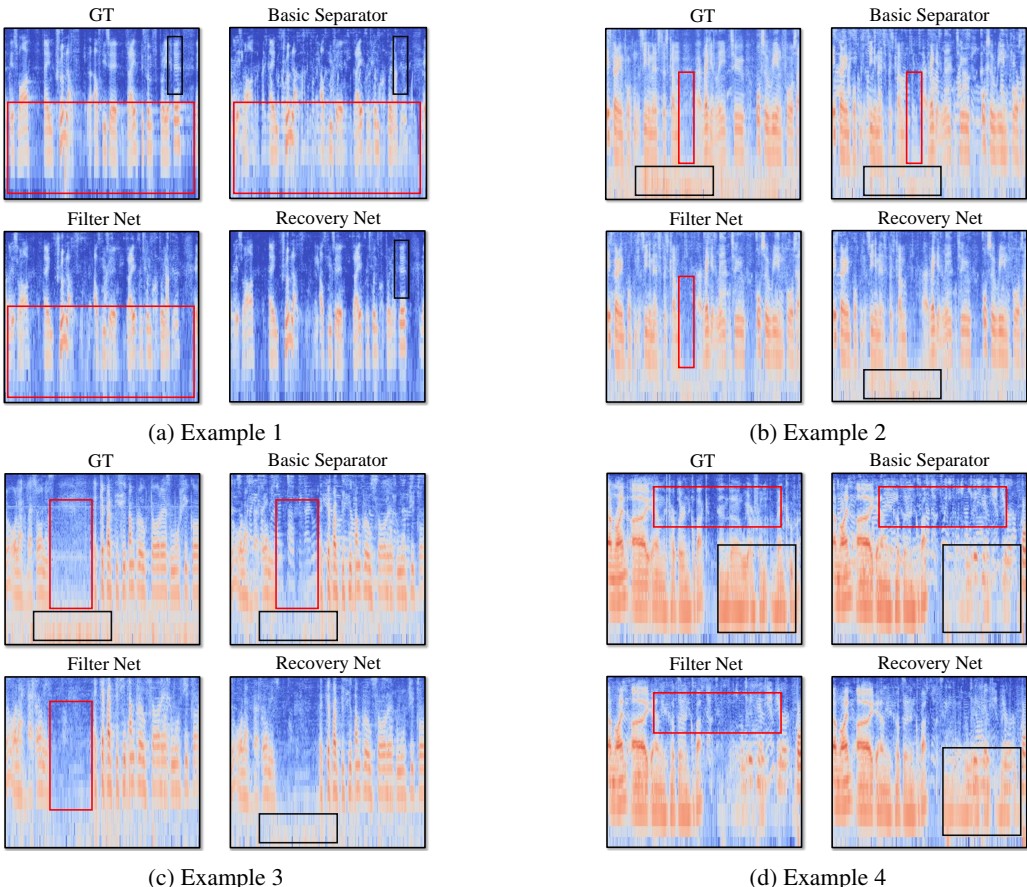

Figure 5: **Visualization of the intensity of spectrogram:** ground truth (GT), outputs of the basic separator, outputs of the Filter Net, and outputs of the Recovery Net.

