# OpenReview forum: "Filter-Recovery Network for Multi-Speaker Audio-Visual Speech Separation"
_ICLR.cc/2023/Conference — ICLR 2023 poster_

### Official Review · Reviewer_t4UV · 2022-10-18

**Confidence:** 4
**Correctness:** 3
**Technical Novelty And Significance:** 3
**Empirical Novelty And Significance:** 2
**Recommendation:** 6

**Clarity, Quality, Novelty And Reproducibility:**

This paper presents the problem and proposed solution quite clearly so that readers may be able to reproduce the results. The proposed model is simple with novelty.

**Strength And Weaknesses:**

This paper first points out that there are two issues in complex multi-speaker separation and solve these two issues with the proposed Filter-Recovery Network (FRNet). The model structure of FRNet is reasonable, but there are still some unclear parts that need to be addressed:

1. The loss function used to train the proposed model is only based on e.q. 13 without any constraints, how do you guarantee the Filter Net and Recovery Net work properly as their original goals (removing noise voices and extracting missing voices, respectively)?
2. Some audio samples should be released in the supplementary material, especially the separated result at each stage: i.e., a) after Basic Audio-Visual Speech Separator, b) after Filter Net and c) after Recovery Network. This can also somewhat solve my first concern.
3. The inputs of the Filter Net are Vi and Mi, why the mixture X doesn’t need to be fed into the model?
4. For the experimental parts, how do you calculate the scores (SDR and PESQ) if there are multiple output-separated speeches?

The proposed method to separate voices for a variable number of speakers with a single model is not very novel, to my understanding, it is mainly based on a proper setting of the training set.

comparison
Typo:
1.	Page 2, Specifically, Specifically, -> Specifically.
2.	Page 6, k-th speaker -> i-th speaker.


**Summary Of The Paper:**

This paper points out that there are two prominent issues in complex multi-speaker separation results: 1) There exist some noisy voice pieces belonging to other speakers; 2) Part of the target speech is missing. A Filter-Recovery Network (FRNet) is hence proposed to solve these problems. The authors also emphasize that their single model can separate voices for a variable number of speakers, which is simply achieved by proper training data setting (i.e., including mixtures with a different number of speakers). Overall, the model design is quite interesting with a good performance improvement.

**Summary Of The Review:**

The prosed method is motivated by two observed problems in the multi-speaker separation. FRNet can also be used as a post-processing module to improve the performance of different audio-visual separation frameworks is a plus. Overall, the paper is well written with good experimental results.

---

> ### Author Response · Authors · 2022-11-18
> **Response to Reviewer t4UV**
>
> 1) The loss function used to train the proposed model is only based on e.q. 13 without any constraints, how do you guarantee the Filter Net and Recovery Net work properly as their original goals (removing noise voices and extracting missing voices, respectively)? Some audio samples should be released in the supplementary material, especially the separated result at each stage: i.e., a) after Basic Audio-Visual Speech Separator, b) after Filter Net and c) after Recovery Network. This can also somewhat solve my first concern.
>
> * Thanks for this advice, we provide several examples to illustrate the effects of Filter Net and Recovery Net in the appendix.
>
> 2) The inputs of the Filter Net are Vi and Mi, why the mixture X doesn’t need to be fed into the model?
>
> * Please refer to the first answer to reviewer 1.
>
> 3) For the experimental parts, how do you calculate the scores (SDR and PESQ) if there are multiple output-separated speeches?
>
> * We report the average metrics for all speakers in the test set. For example, there are a total of 7200 speakers in the VoxCeleb2 unseen 2-mix set, so the metrics are the average performance of 7200 speakers. We modify the caption of Table 2 to give this description.
>
> 4) The proposed method to separate voices for a variable number of speakers with a single model is not very novel, to my understanding, it is mainly based on a proper setting of the training set.
>
> * We indeed propose a training setting that simultaneously deals with different kinds of mixtures, expecting to better separate speeches in more complex scenes in a single model.

---

### Official Review · Reviewer_VNdU · 2022-10-21

**Confidence:** 4
**Correctness:** 3
**Technical Novelty And Significance:** 2
**Empirical Novelty And Significance:** 2
**Recommendation:** 6

**Clarity, Quality, Novelty And Reproducibility:**

It is easy to follow the paper but proofreading is needed. A non-exhaustive list of typos from the introduction  is the following (but can be found in all sections):
Firstly, Filter module -> Firstly, the filter module
the Recovery module use -> uses
most of works -> most works
Specifically, Specifically -> specifically

Fig. 2 is a bit confusing. It’s only understandable after reading the text. It would be better if all the necessary information to fully understand it is contained in the figure or caption.

The main novelty is the introduction of the FRnet. It is shown that if combined with other existing speech separation networks leads to improved performance.

It is not possible to reproduce all the results without help from the authors. The test sets are generated by random sampling.


**Details Of Ethics Concerns:**

No concerns.

**Strength And Weaknesses:**

Strengths

Several results and an ablation study is presented which convincingly show that the proposed approach outperforms other existing approaches.

The proposed FRNet can improve the performance of other speech separation approaches and this is a nice contribution.

Weaknesses

First of all, writing can be improved. Although the text is understandable there are several errors/typos. More details in the next section.

The impact of facenet/lipnet is missing in the ablation study. It would be good to show how important each of them is. Since most of the information is contained in the lip movements the contribution of facenet might be small.

The authors emphasise that one of the main contributions is that the proposed method can separate mixtures with a variable number of speakers simultaneously during training. Isn’t this something the existing approaches can already do? And also the number of speakers needs to be known in advance. It’s not clear why this is an important contribution.

It is not explained why the authors chose a two-step approach instead of an end-to-end approach, i.e., why not integrating the filter and recovery networks in the speech separation model and optimise them jointly?

The proposed audio-visual speech separation network (without the FRNet) is very similar to other existing audio-visual approaches. Why is this model better? This is also not clear.

Some details are not explained, e.g. the lambda coefficient in eq. 13 is set to 0.5, how is this value chosen? Why is training performed for 19 epochs? Why 2.55 seconds are used for training? It seems these numbers have been chosen via an optimisation stage but it is not explained how this was performed.


**Summary Of The Paper:**

This work presents an audio-visual speaker separation method suitable for multi-speaker scenarios. An audio-visual speaker separator is first introduced which is followed by a filter-recovery network which aims to further improve the separation quality. The latter can also be combined with other separator networks and it is shown that it can improve their performance as well. The proposed approach is tested on standard benchmarks and it  outperforms other state-of-the-art methods.

**Summary Of The Review:**

Overall, this is an interesting contribution but there are several weakness as explained above.

---

> ### Author Response · Authors · 2022-11-18
> **Response to Reviewer VNdU**
>
> 1) First of all, writing can be improved.
>
> * We greatly appreciate your suggestion. We have carefully checked the paper and corrected the mistakes we found.
>
> 2) The impact of facenet/lipnet is missing in the ablation study.
>
> * Thanks for this suggestion. We conduct an ablation study in Table 4 to explore the effect of individual visual nets. As expected, both types of visual signals are beneficial. Due to the high relevance of lip motion and continuous pronunciation, lip information plays a more critical role than static face image.
>
> 3) The authors emphasise that one of the main contributions is that the proposed method can separate mixtures with a variable number of speakers simultaneously during training. Isn’t this something the existing approaches can already do? And also the number of speakers needs to be known in advance.
>
> * As far as we know, most multi-speaker speech separation methods are designed for a fixed number of speakers, mostly 2 or 3 speakers. Some methods train separate models for each kind of mixture with various numbers of speakers, such as [Audio-Visual Speech Enhancement Method Conditioned on the Lip Motion and Speaker-Discriminative Embeddings; Looking to Listen at the Cocktail Party: A Speaker-Independent Audio-Visual Model for Speech Separation]. The most similar method to ours is [The Conversation: Deep Audio-Visual Speech Enhancement], but it focuses on the speech enhancement task.
>
> 4) It is not explained why the authors chose a two-step approach instead of an end-to-end approach.
>
> * There may be some unclear descriptions in the paper. Our method is indeed an end-to-end approach that trains the basic separator and FRNet jointly by optimizing the outputs of the basic separator and FRNet simultaneously. We have revised the relevant descriptions in the paper to make it unmistakable.
>
> 5) The proposed audio-visual speech separation network (without the FRNet) is very similar to other existing audio-visual approaches. Why is this model better?
>
> * The basic separator we adopt is VisualVoice with a slight modification. We change the simple audio-visual feature concatenation in VisualVoice to the AV-Fusion module proposed in [Audio-visual speech separation based on joint feature representation with cross-modal attention], which performs audio-visual cross attention to obtain an enhanced feature. Our basic separator achieves better results in VoxCeleb2 test sets. In fact, we do not emphasize our contribution to the basic separator, and only regard it as one of the choices, which can be replaced by any network with same input and output.
>
> 6) Some details are not explained, e.g. the lambda coefficient in eq. 13 is set to 0.5, how is this value chosen? Why is training performed for 19 epochs? Why 2.55 seconds are used for training? It seems these numbers have been chosen via an optimisation stage but it is not explained how this was performed.
>
> * Thanks for your advice. 1) Empirically, we set the lambda in eq.13 to 0.5 to balance the training of the basic separator and FRNet. In addition, we adopt a larger lambda 0.8 (mainly optimizing the basic separator) and a smaller lambda value 0.2 (focusing on optimizing FRNet). The results can be referred to Table 8 in the appendix. We find that attaching a higher weight to the training loss of FRNet (lambda 0.2) leads to minor differences from equal loss weights. However, when we reduce the loss weight of FRNet (lambda is 0.8), it results in a non-negligible decrease in performance, which also proves the importance of FRNet. 2) We observe that the training loss curve almost does not drop after training 19 epochs, so we optimize the models for 19 epochs. 3) Following [VISUALVOICE: Audio-Visual Speech Separation with Cross-Modal Consistency], we randomly take 2.55s long visual clips and the corresponding audios from videos, which is appropriate for training a batch of data. We add some explanations in the related sections in the paper.
>
> 7) It is easy to follow the paper but proofreading is needed. A non-exhaustive list of typos from the introduction is the following (but can be found in all sections): Firstly, Filter module -> Firstly, the filter module the Recovery module use -> uses most of works -> most works Specifically, Specifically -> specifically
>
> * Thanks for giving the advice. We have thoroughly checked the paper and rectified the errors as much as possible.
>
> 8) Fig. 2 is a bit confusing. It’s only understandable after reading the text. It would be better if all the necessary information to fully understand it is contained in the figure or caption.
>
> * We modify Figure 2 and add a more detailed description in the caption to make it clearer.
>
> 9) It is not possible to reproduce all the results without help from the authors. The test sets are generated by random sampling.
>
> * We will release the data list of test sets along with the code.

---

> > ### Comment · Reviewer_VNdU · 2022-11-28
> > **Reply**
> >
> > The authors have addressed most of my concerns. Explanations for points 3,5, 6 above should be included in the main manuscript or appendix.

---

> > > ### Author Response · Authors · 2022-11-29
> > > **Response to the reply**
> > >
> > > Thanks very much for the suggestions and kind support of this work. Your constructive feedback and criticisms help us greatly towards improving this work. We will add the discussed details in our final version.

---

### Official Review · Reviewer_X56m · 2022-10-23

**Confidence:** 3
**Correctness:** 3
**Technical Novelty And Significance:** 3
**Empirical Novelty And Significance:** 1
**Recommendation:** 6

**Clarity, Quality, Novelty And Reproducibility:**

- Notation can be improved
  - The authors describe that “[we] feed $V_i$ and $Au_i$ to AV-Fusion module to obtain a[n] enhanced feature.” Why is the audio feature $Au_i$ speaker dependent? According to the diagram it is the output from the encoder that takes mixture spectrogram X as input.
  - In Sec 3.3 Recovery Net, why are M_{1, \cdots, S} referred to as “coarse” masks? Do they have different temporal or frequency resolution?
  - The paper can use more proofreading. There are still some typos.
- Experiments are good but can be improved as suggested in the Weakness section.
- The idea of refining masks for source separation is novel
- The proposed method appears to easy to reproduce



**Strength And Weaknesses:**

Strengths

- BFRNet yields better performance on VoxCeleb2, LRS2, LRS3 compared to prior works
- The mask refinement module, FRNet, is complementary to other masked-based audio-visual speech separation models according to the experiments in Table 2 and 3.
- Ablation studies confirm that both the filter and the recovery module contribute to the improvement. It also shows video facilitates the filter network, and using the cleaned mask to the recovery module improves.

Weaknesses
- While the empirical results are strong, this paper can be improved by providing more explanations of why the FRNet further improves the base audio-visual speech separation module.
  - Why does the filter network remove the non-target speakers’ voice better than the base separation module? The filter network only takes predicted mask and video features as the input. It does not even take the mixture audio as the input. It is surprising that it can tell where the noise is by just looking at a mask.
  - Similarly, why does the recovery network learn what to keep better than the base audio-visual source separation network?
  - It is possible that adding FRNet improves because now the entire model is bigger than the one without. To justify the FRNet carries benefits other than increasing the model capacity, the authors are recommended to compare with a stronger base model (e.g., Basic Audio-Visual Speech Separator) that have a deeper encoder/decoder where the parameter counts match roughly that of the BFRNet.


**Summary Of The Paper:**

This paper studies masking-based audio-visual source separation, which predicts a complex spectral mask for the audio mixture for each speaker conditioning on the mixture speech and the video of the target speaker. The authors proposed BFRNet, which is composed of a audio-visual source separation model that predicts a mask for each speaker, and a filter-and-recovery network (FRNet) that refines predicted masks to a) remove non-target residual speech and b) to recover target speech removed from the initially predicted mask.

**Summary Of The Review:**

Experimental results are strong compared to the prior work. The main novelty is a module for mask refinement, which can improve several existing audio-visual speech separation models as demonstrated by the authors. More controlled experiments are required to justify the gain does not simply result from the increase in the number of parameters.

---

> ### Author Response · Authors · 2022-11-18
> **Response to Reviewer X56m**
>
> 1) Why does the filter network remove the non-target speakers’ voice better than the base separation module? The filter network only takes predicted mask and video features as the input. It does not even take the mixture audio as the input. It is surprising that it can tell where the noise is by just looking at a mask. Similarly, why does the recovery network learn what to keep better than the base audio-visual source separation network?
>
> * It's easier for FRNet to obtain more accurate results than the basic separator because the former solves a less complex problem than the latter. The input masks to FRNet are the outputs of the basic separator generated from the enhanced audio-visual features, which incorporate the feature of the mixture spectrogram. Therefore, the Filter Net implicitly utilizes the mixture spectrogram. For Recovery Net, since the input masks are based on the same mixture spectrogram, the masks manifest the same pattern as the spectrograms. Therefore, the learned patterns from spectrograms are equivalent to that from masks.
>
> 2) It is possible that adding FRNet improves because now the entire model is bigger than the one without. To justify the FRNet carries benefits other than increasing the model capacity, the authors are recommended to compare with a stronger base model (e.g., Basic Audio-Visual Speech Separator) that have a deeper encoder/decoder where the parameter counts match roughly that of the BFRNet.
>
> * We conduct experiments with a deeper basic separator with nearly the same capacity as BFRNet. (The former is 290Mb, and the latter is 293Mb). The experimental results of this model can referred to in the "DeBase" method in Table 2 and Table 3 in Section 4.3. The results show that only increasing the model capacity cannot effectively improve performance.
>
> 3) The authors describe that “[we] feed Vi and Aui to AV-Fusion module to obtain a[n] enhanced feature.” Why is the audio feature Aui speaker dependent? According to the diagram it is the output from the encoder that takes mixture spectrogram X as input.
>
> * Thanks for pointing out this error. We denote the audio feature of mixture spectrogram X as Aui by mistake, and indeed it is Au. We have corrected this error in the corresponding place of the paper.
>
> 4) In Sec 3.3 Recovery Net, why are M_{1, \cdots, S} referred to as “coarse” masks? Do they have different temporal or frequency resolution?
>
> * The description of the corresponding Section may be ambiguous. We mean "coarse" masks as the separation results of the basic audio-visual speech separator, which are used to distinguish the refined masks obtained by FRNet.
>
> 5) The paper can use more proofreading. There are still some typos.
>
> * Thank you for giving this advice. We have double-checked the writing and corrected the mistakes we found.

---

> > ### Comment · Reviewer_X56m · 2022-12-02
> > **Reply to the response**
> >
> > The authors have addressed most of my questions. I'm raising my overall rating to 6.

---

### Decision · Program_Chairs · 2023-01-20

**Decision:**

Accept: poster

**Justification For Why Not Higher Score:**

All three reviewers ranked the paper as "marginally above the acceptance threshold", and this matches the paper quality and contribution.

**Justification For Why Not Lower Score:**

This paper addresses an interesting problem and the proposed method is shown to be effective.

**Metareview: Summary, Strengths And Weaknesses:**

The paper is concerned with separating speech signals in a multi-speaker mixture and outputting the speech signals corresponding to each given face. In particular, the authors propose a method called BFRNet consisting of a basic separation module and a Filter-Recovery Network (FRNet). The FRNet filters the noisy speech and recovers the missing parts in the separation results of the basic separator. Experiments were conducted to valid the effectiveness of the proposed method.

**Note From Pc:**

if the above contains the word "oral" or "spotlight" please see: "oral" presentation means -> notable-top-5% and "spotlight" means -> notable-top-25%. As stated in our emails, we are disassociating presentation type from AC recommendations

**Summary Of Ac-Reviewer Meeting:**

A meeting was held among reviewers and AC. The main points are 1) the proposed method has moderate novelty, 2) the experiments including ablation studies are extensive and 3) the reviewing and rebuttal process cleared several concerns and strengthened the paper. Overall, this is a borderline paper.